# The Aging Stress Response and Its Implication for AMD Pathogenesis

**DOI:** 10.3390/ijms21228840

**Published:** 2020-11-22

**Authors:** Janusz Blasiak, Elzbieta Pawlowska, Anna Sobczuk, Joanna Szczepanska, Kai Kaarniranta

**Affiliations:** 1Department of Molecular Genetics, Faculty of Biology and Environmental Protection, University of Lodz, 90-236 Lodz, Poland; 2Department of Orthodontics, Medical University of Lodz, 92-216 Lodz, Poland; elzbieta.pawlowska@umed.lodz.pl; 3Department of Gynaecology and Obstetrics, Medical University of Lodz, 93-338 Lodz, Poland; anna.sobczuk@umed.lodz.pl; 4Department of Pediatric Dentistry, Medical University of Lodz, 92-216 Lodz, Poland; joanna.szczepanska@umed.lodz.pl; 5Department of Ophthalmology, University of Eastern Finland, 70211 Kuopio, Finland; kai.kaarniranta@kuh.fi; 6Department of Ophthalmology, Kuopio University Hospital, 70211 Kuopio, Finland

**Keywords:** the aging stress response, aging, age-related macular degeneration, AMD, insulin/IGF-1, SIRT1, PGC-1α, autophagy, DNA damage response, mitochondrial quality control

## Abstract

Aging induces several stress response pathways to counterbalance detrimental changes associated with this process. These pathways include nutrient signaling, proteostasis, mitochondrial quality control and DNA damage response. At the cellular level, these pathways are controlled by evolutionarily conserved signaling molecules, such as 5’AMP-activated protein kinase (AMPK), mechanistic target of rapamycin (mTOR), insulin/insulin-like growth factor 1 (IGF-1) and sirtuins, including SIRT1. Peroxisome proliferation-activated receptor coactivator 1 alpha (PGC-1α), encoded by the *PPARGC1A* gene, playing an important role in antioxidant defense and mitochondrial biogenesis, may interact with these molecules influencing lifespan and general fitness. Perturbation in the aging stress response may lead to aging-related disorders, including age-related macular degeneration (AMD), the main reason for vision loss in the elderly. This is supported by studies showing an important role of disturbances in mitochondrial metabolism, DDR and autophagy in AMD pathogenesis. In addition, disturbed expression of PGC-1α was shown to associate with AMD. Therefore, the aging stress response may be critical for AMD pathogenesis, and further studies are needed to precisely determine mechanisms underlying its role in AMD. These studies can include research on retinal cells produced from pluripotent stem cells obtained from AMD donors with the mutations, either native or engineered, in the critical genes for the aging stress response, including *AMPK*, *IGF1*, *MTOR*, *SIRT1* and *PPARGC1A*.

## 1. Introduction

The biological fitness of the human organism is supported by the phylogenetically conserved stress response, maintenance and repair pathways. They include nutrients and energy signaling, proteostasis, DNA damage response (DDR), mitochondrial quality control (mtQC) and enzymes metabolizing toxic species [1]. The efficacy of these pathways declines with aging, but the process of aging is also associated with diverse stresses that impose signals coming from the environment (reviewed in [2]). Therefore, an aging organism faces increased stress, which should be counterbalanced by the aging stress response with signaling pathways to maintain cellular and organismal homeostasis (reviewed in [3]). The impairment of the aging stress response may contribute to syndromes associated with premature aging, age-related diseases and cancer.

Age-related macular degeneration (AMD) is the primary cause of legal blindness in the elderly in developed countries. Its estimated prevalence in the world is 196 million in 2020 and is projected to increase to 288 million by 2040. Despite these numbers, no effective treatment is available, except for some specific cases of the neovascular (wet) form of AMD (reviewed in [4]). However, the treatment in those cases is mainly aimed at vision loss prevention. This limitation of effective therapeutic options is likely due to highly incomplete knowledge of mechanism(s) involved in AMD pathogenesis. It is commonly accepted that aging is the most serious risk factor in AMD. Therefore, aging-related stress and response to this stress may contribute to AMD pathogenesis. Oxidative stress is often considered a major risk factor in AMD [5]. It is associated with increased levels of reactive oxygen and nitrogen species (ROS and RNS) that may damage cellular structures and macromolecules, including proteins and DNA. This may result in misfolded and dysfunctional proteins and consequently dysfunctional subcellular structures and organelles. These proteins should be removed and recycled by proteostasis, which dynamically regulates the proteome to keep it balanced and functional and is an essential pathway of the stress response [6]. Damage to DNA induces DDR, another stress response component [7]. However, oxidative stress and increased ROS/RNS levels are associated with several other AMD risk factors, including improper diet, tobacco smoking, blue light exposure and others [8]. Moreover, oxidative stress is common in the retina as it is the tissue of high consumption of oxygen, contains a high proportion of polyunsaturated fatty acids and is exposed to visible light. Therefore, oxidative stress is also present in young retinas that are generally not susceptible to AMD, and there must be an additional factor(s) associated with age that superimposes the stress in older individuals. There are substantial differences in everyday life between young and older adults, such as nutrient intake and physical activity that may influence AMD susceptibility [9]. The aging stress response seems to be a prime candidate to distinguish AMD susceptibility of younger and older adults as it combats many AMD risk factors and is declined with aging.

In this review, we present and update information on the aging stress response pathways associated with nutrient signaling, DDR, proteostasis (mainly autophagy) and mtQC. Then, we show how these pathways are affected in AMD patients or models to justify that the aging stress response may be the key element in AMD pathogenesis.

## 2. The Aging Stress Response

Aging is frequently understood as the process resulting from the accumulation of detrimental biological changes over time, which increase an organism’s susceptibility to disease and make it more likely to die. However, the causal relationship between the biological changes that occur with time (biological aging) and metrical (chronological) aging is far from understanding [10].

Aging may be considered as a regulated process controlled by some conserved signaling pathways, including the insulin/IGF-1 (insulin-like growth factor 1), mTOR (mechanistic target of rapamycin), AMPK (5’AMP-activated protein kinase) and sirtuins [11]. These pathways detect stress-related events and activate proteins of adaptive cellular responses that reduce the consequences of environmental insults and aging-related stress [3]. These adaptive responses include DNA repair, proteostasis with autophagy and mitochondrial quality control—all of them have been reported impaired in AMD patients and AMD cellular and animal models [12,13,14,15,16].

### 2.1. Nutrient-Signaling Pathways in Aging

Nutrient signaling plays an important role in the control of aging and age-related disorders, and it was the first pathway to be demonstrated to do so in model organisms [17]. Caloric restriction, one of the best-known way of lifespan extension, acts through multiple signaling pathways, including the insulin/IGF-1, mTOR and AMPK pathways and sirtuins [18].

In an aging organism, the level of active IGF-1 decreases, which has been associated with human frailty and cognitive decline [19]. Mutations in the genes of the insulin/IGF-1 pathway are reported to prolong the life of model organisms [20]. This pathway is a molecular connection between dietary intake and the cellular stress response pathway, so it can be modulated by a dietary intervention. Aging and age-related pathologies relate to the metabolic status of sirtuins, a family of highly conserved proteins [21,22]. They display NAD^+^ (nicotinamide adenine dinucleotide)-dependent protein deacetylase activity, and this is NAD^+^-sensing that determines their association with metabolic status [23]. Among seven mammalian sirtuins, SIRT1 is regulated by nutrient status in the cell, triggers stress response pathways and induces changes in energy metabolism [24]. The expression and activity of SIRT1 decrease with age and a diet rich in fat [25]. On the other hand, its activity increases in nutrients deficiency, including caloric restriction [26]. Active SIRT1 deacetylases several substrates that are involved in aging and stress response, including peroxisome proliferation-activated receptor coactivator 1 alpha (PGC-1α) [27]. Numerous studies suggest that SIRT1 may be involved in the extension of lifespan in stress conditions and can do so through the regulation of stress–response processes, including DNA repair, mitochondria and protein quality control as well as cell survival [21] (Figure 1). It is still an open question whether it can also do so in normal conditions.

A low status of cellular energy, which can be reflected by a low ATP/AMP ratio, may activate AMPK that is not only a sensor of such status but also becomes an energy reservoir [28]. The stress of a low-energy state evokes an increase in catabolic pathways promoted by signaling pathways transcriptionally and post-transcriptionally activated by AMPK [29,30]. AMPK promotes the switch from fatty acid synthesis to oxidation during nutritional stress resulting in low ATP level [3].

Another nutrient sensor, mTOR, integrates environmental signals, including those coming from nutrients, to control growth and metabolism in eukaryotic cells [31]. Nutrient limitation decreases mTOR, and its activity is suppressed in models of longevity (reviewed in [32]). Moreover, its inhibition extended lifespan in various model organisms [33]. In mammals, mTOR acts in two separate signaling complexes, mTORC1 and mTORC2 and their age-related effects are mainly underlined by modulation of protein synthesis and autophagy [34].

### 2.2. Mitochondria in Aging

Although the mitochondrial theory of aging, originating from Harman’s “free radical theory of aging”, is no longer a paradigm, mitochondria are still a central element in the process of aging [35]. Such a prominent role is supported by research with overexpression of human catalase targeted to mouse mitochondria and extending lifespan of transgenic animals by about five months as compared with three months or one month for the enzyme targeted the peroxisome or the nucleus, respectively [36]. The signs of aging: cardiac pathology and cataract development were delayed in animals with catalase targeted to mitochondria. Moreover, the most pronounced antioxidant effect was observed for mtDNA, suggesting that aging may be linked with the production of ROS in mitochondria [37]. Mice with human catalase targeted to mitochondria were also exploited in models of various diseases, including metabolic syndrome and atherosclerosis, cardiac aging, heart failure, skeletal muscle pathology, sensory defect, neurodegenerative diseases and cancer, suggesting that mitochondrial production of ROS and damage to mitochondria may play an essential role in the process of aging and aging-related pathologies [38].

Studies on other mouse models, including animals with mutations in the gene encoding polymerase gamma, the mitochondrial replicase, showed that the association between aging and mitochondrial ROS production and mitochondrial homeostasis was complex [39,40,41]. These studies also pointed to the role of apoptosis in aging. Mutations in mtDNA may either extend or reduce the lifespan of model organisms, but the lack of mtDNA in yeast (“the petit mutation”) is linked with lifespan extension [42]. As was later shown, this effect was attributed to the lack of mtDNA per se and not to changes in respiration and reduced ROS production [43].

Mitochondrial quality control is accomplished by the coordination of mitochondrial biogenesis, dynamics and mitophagy, a specialized pathway of autophagy [44] (Figure 2). Mitochondrial transcription factors, including mitochondrial transcription factor A (TFAM), are encoded by nuclear genes and transported to mitochondria to target mtDNA. TFAM upregulates genes encoding the mitochondrial electron transport chain (mtETC), resulting in an elevated oxygen intake, ATP production and mitochondrial content. Mitochondrial turnover is mainly controlled by fusion (mitofusin 1 (MFN1), MFN2 and mitochondrial dynamin-like 120 kDa protein (OPA1)) and fission (dynamin 1-like protein (DNM1L) and mitochondrial fission 1 protein (FIS1)) proteins [45,46]. Damaged mitochondrial components can be degraded and recycled by mitophagy [47].

Several pathways and proteins control mitochondrial response to stress associated with low energy allowing mitochondria to adjust their energy usage and the number of organelles [3]. These include PGC-1α, sirtuins, mTOR and AMPK. Therefore, mitochondrial response to age-related stress crosstalk with nutrition signaling. PGC-1α is a key regulator of mitochondrial biogenesis and an important factor in other aspects of mtQC [42]. Moreover, emerging evidence suggests that PGC-1α is a major regulator of lifespan and its interaction with SIRT1, AMPK, and mTOR is important for this function [48,49,50,51].

A mild mitochondrial impairment may activate a stress response pathway, which promotes a protective milieu counterbalancing a potentially shorter lifespan associated with the mitochondrial impairment [3]. Whether it is a classical hormesis response or results from the interaction of PGC-1α with SIRT1, AMPK and mTOR, remains to be established, although mTOR seems to be a prime candidate to link mitochondrial biogenesis and cellular metabolism with lifespan [3].

### 2.3. DDR in Aging

Increasing DNA damage resulting from the aging process is predicted by all theories of aging [52]. Moreover, not only DNA but other cellular macromolecules undergo damage in aging [53]. Unrepaired DNA damage may be changed into mutation (Figure 3). Therefore, aging is associated with accumulating damage to cellular molecules that may become non-functional. This may contribute to age-related degenerative diseases, and increased levels of DNA damage and mutations lead to genomic instability, associated with almost all, if not all, cancers [54,55,56]. In addition, the epigenetic profile changes with aging, which reflects a complex relationship between aging and epigenetics (reviewed in [57]). Many aspects of the nutritional signaling in aging, including longevity-related interactions of AMPK, mTOR, SIRT1 and PGC-1α, are underlined by changes in the epigenetic profile [58].

Aging-related DNA damage induces DDR, which may be impaired from the very start as potentially each of its proteins can be damaged in aging. In fact, many reports suggest that the capacity of DNA repair, the main component of DDR, declines with age, which contributes to both increased DNA damage and genomic instability [52,59,60,61,62].

Although we are far from the generally accepted definition of aging, premature aging is likely commonly understood. In humans, premature aging and early death are associated with defects in DNA repair or, more generally—in DDR [63,64,65,66,67]. However, several studies report not only an association but also a causal relationship between DNA repair defects and premature aging [68,69,70,71,72]. Werner syndrome is likely the most characterized human premature aging disorder, which is closely associated with defects in DNA repair as this syndrome is caused by recessive mutations in the *RecQ* gene, encoding a DNA helicase involved in DNA repair [73].

Studies of Werner syndrome and other human progeroid disorders suggest an association between DDR and signaling pathways involved in the aging process [3]. Mutations in the essential DNA repair genes *XPF-ERCC1* (ERCC4 excision repair 4, endonuclease catalytic subunit) and *TP53* (tumor protein p53) resulted in downregulation of the insulin/IGF pathway in cells from progeroid syndrome patients and mice [74,75].

Another example of the association of DDR with aging is ataxia–telangiectasia (AT), underlined by mutations in the *ATM* (ATM serine/threonine kinase, ataxia telangiectasia mutated) gene [76]. The ATM protein, along with ATR (ATR serine/threonine kinase, ataxia telangiectasia and Rad3-related protein), is the central regulator of DDR, responsible for the transduction of the signal coming from DDR sensors, recognizing damaged DNA or disturbed DNA replication and activating DDR effectors, controlling cellular processes aimed at restoring the DNA structure or resuming DNA replication [77]. In general, ATM and ATR have a broad spectrum of substrates, including proteins playing an important role in signaling pathways regulating aging [78].

DNA damage response may not only restore the integrity and fidelity of DNA, but it may induce senescence or programmed cell death, which may contribute to aging phenotype [79]. Moreover, some DNA damages may be tolerated, depending on the cellular context. In general, when DNA damage does not exceed the cell repair capacity, it is repaired, but when the extent of the damage is above such capacity, the cell cycle is arrested for giving the cell additional time for repair. If this fails, the cell becomes senescent or is directed to a programmable death pathway, usually apoptosis. Therefore, DDR may differ in mitotic and postmitotic cells, and this is likely responsible for different AT phenotypes in different tissues: loss of ATM-mediated signaling in mitotic cells results in cancer(s), but in postmitotic neurons—in degeneration [80,81]. In addition, in mitotic cells, DDR may tolerate or misrepair some DNA damages to preserve the process of replication. Therefore, genotoxic stress induces a response with many pathways that may be affected by aging, leading to enhancing aging-related phenotypes, including aging-related disease and death.

DNA damage response in mitochondria (mtDDR), although not fully known, is likely poorer than its nuclear counterpart. mtDDR does not directly induce apoptosis but may lead to the degradation of highly damaged mtDNA. Several aspects of aging-related mtDNA are directly or indirectly associated with mechanisms of mtQC, described in the previous section.

### 2.4. Autophagy and Aging

Proteostasis, a process ensuring proper protein folding, sequestering of misfolded proteins, degradation of damaged or no longer needed proteins, is performed by the collective action of a set of molecular chaperones, transcription factors and cofactors.

Proper protein folding is ensured by UPR, improperly folded proteins are sequestered in inclusion bodies, and degradation occurs in the proteasome and in autophagy (Figure 4). Proteostasis is declined with age, which may result in the escape of misfolded proteins from sequestration and non-functional/damaged proteins from degradation [82]. Both misfolded and damaged proteins may be dysfunctional, further contributing to deleterious changes, increasing susceptibility of the organism to disease, and making it more likely to die.

Autophagy declines with aging, contributing to the accumulation of dysfunctional organelles and proteins as well as their aggregation [83]. Conversely, efficient autophagy may contribute to lifespan extension, as it was shown in research on model organisms (reviewed in [84]). However, few studies show that autophagy alone is sufficient to extend lifespan—instead, it is required for life extension induced by caloric restriction or reduced insulin signaling [85]. It seems that autophagy may take a special position in proteostasis as it may be involved in many effects, phenomena and systems, including oxidative stress defense, DDR and perform the pro-death and pro-survival functions [86]. Therefore, autophagy may contribute to the regulation of lifespan on many distinct pathways.

Mammalian studies on autophagy confirm research on lower organisms and demonstrate an essential role of autophagy in aging and age-related neurodegeneration [87]. Moreover, autophagy, due to its multifaceted nature, may play a role in many aspects of age-related stress responses. Autophagy activation is closely associated with nutrient status and various other stimuli, such as energy stress, endoplasmic reticulum (ER) stress, pathogen- and danger-associated molecular patterns, hypoxia, redox stress and mitochondrial damage [88]. During starvation, autophagy provides cell recycled amino acids for protein synthesis.

Osteoarthritis is the most common form of arthritis and can be considered as a sign of aging because aging is its primary risk factor [89]. It was shown that SIRT1 was decreased in human normal aged and osteoarthritis cartilage compared with young cartilage [90]. Moreover, activation of SIRT1 increased autophagy in chondrocytes by the deacetylation of essential autophagy proteins, including Beclin1, ATG5 (autophagy-related 5), ATG7, LC3 (MAP1LC3A, microtubule-associated protein 1 light chain 3 alpha) in an mTOR/ULK1 (Unc-51-like autophagy activating kinase 1)-independent manner. These results confirm an important role of SIRT1 in aging-related stress response through the involvement in the autophagic pathway.

Mitochondria-specific autophagy, or mitophagy, also declines with age, and its dysfunction may further potentiate an age-related phenotype [91]. In line with this statement is the observation that old mice fed with a mitophagy inducer showed an improvement in age-related cardiac functions and increased levels of mtQC proteins, including PARKIN, BNIP3 (BCL2 interacting protein 3), SIRT3 and PGC-1α [92].

## 3. Age-Related Macular Degeneration—A Disease of Aging Stress Response

AMD affects the macula, a small structure in the center of the retina, responsible for sharp and color vision.

Clinically, advanced AMD may be divided into two forms: dry (atrophic) and wet (neovascular) AMD (Figure 5). Dry AMD is characterized by the presence of drusen, small white or yellowish objects that are made of protein and lipids. Dry AMD may lead to vision loss as the development of this disease to geographic atrophy (GA) may impair the functionality of photoreceptors and eventually cause their death along with underlying retinal pigment epithelium (RPE) cells atrophy [93,94]. In wet AMD, the RPE induces the synthesis of angiogenic factors that stimulate the production of new blood vessels from supporting choriocapillaris (choroidal neovascularization, CNV). These vessels penetrate the Bruch’s membrane-RPE complex leading to photoreceptor death. In some cases, large hemorrhages develop in the subretinal area.

Although AMD pathogenesis is not fully known, many factors, both genetic/epigenetic and environmental/lifestyle, may be involved [95]. Mechanisms and factors of AMD pathogenesis are described in many excellent reviews (for example, [96,97,98,99]). In this review, we focus on the elements of AMD pathophysiology directly related to aging and the aging stress response: autophagy as a pathway of proteostasis, DDR and mtQC.

As mentioned, aging is likely the most serious risk factor for AMD, and it induces disturbances in signaling pathways, which are important in AMD. These changes can be potentiated by other environmental/lifestyle factors. For instance, aging evokes detrimental changes in mitochondria that may lead to ROS overproduction, damage to mtDNA and synthesis of faulty proteins of mtETC complex [100]. Damaged mtETC produces more ROS than in normal conditions. However, a similar effect can be obtained by environmental factors damaging mtDNA. This strong dependence of AMD pathogenesis on aging is supported by the presence of increased levels of amyloid in AMD retinas, suggesting that AMD may be a kind of dementia of the eye [101,102]. Many studies report an enhanced release of the brain amyloid beta 1–42 (Aβ) in stress conditions [103]. Moreover, the production and clearance of Aβ are reported to depend on stress [104]. Therefore, the deposition of Aβ may result from a decline of the stress response with aging. In summary, amyloid beta deposits observed in AMD retinas may be associated with a declined aging stress response.

To date, more than half of the AMD genomic heritability can be explained by both common and rare genetic variants in 34 genetic loci, which are mostly involved in the complement system, extracellular matrix remodeling and lipid metabolism [105]. A common variant (rs1061170) in the complement factor H (*CFH*) has been identified by several research groups to strongly influence AMD risk [106]. The age-related maculopathy susceptibility 2 and high-temperature requirement A serine peptidase 1 (*ARMS2/HTRA1*) is another locus reported to be strongly involved in AMD pathogenesis [107].

### 3.1. Nutrient Signaling in AMD

Some studies limit micronutrients in AMD prevention and therapy to supplementation with vitamin C and E as well as the carotenoids lutein and zeaxanthin (reviewed in [9]). In addition, dietary zinc and copper are recommended in the Age-Related Eye Disease Study (AREDS) and AREDS2 [108]. These recommendations are mainly based on the pivotal role of oxidative stress in AMD pathogenesis as the recommended substances are low molecular-weight antioxidants. However, we have recently suggested that the beneficial effect of zinc in AMD may be related to the amelioration of autophagy, disturbed in the course of the disease [109]. In addition, the Mediterranean diet was recommended to decrease the prevalence of early AMD in the Carotenoids Age-Related Eye Disease Study (CAREDS) [110]. Adherence to such a diet was positively correlated with a lower incidence of AMD in several other studies [9,111,112]. A Mediterranean-type diet is rich in plant foods (fruits, nuts, legumes and cereals) and fish, with olive oil as the dominating fat and moderate amount of wine and low amount of red meat and poultry. It was generally observed that the closer adherence to this kind of diet slows the progression of early AMD to its advanced form and to larger drusen, and this relationship was modulated by the CFH genotype, suggesting that the biological mechanism behind the protective action of Mediterranean-type diet is involved with the complement system.

Fatty acids, including omega-3 long-chain polyunsaturated fatty acids (omega-3 LC-PUFAs), eicosapentaenoic acid and docosahexaenoic acid, are reported to reduce the risk for AMD (reviewed in [113]). Although biochemical and epidemiological studies suggest a high potential of fatty acids in AMD, the molecular mechanism of their involvement in AMD pathogenesis and specifically drusen formation is poorly known [114]. Despite the promising results from dietary interventions with fatty acids, it is still not clear whether the fatty acids that enter the body through the diet or their counterpart produced locally by specific cells are to be the target for dietary intervention. Fatty acids are consumed along with other nutrients that could affect lipid metabolism, which likely results that the Mediterranean-type diet was reported to be more effective than a diet supplemented with just lipids [115].

Golestaneh’s lab addressed the problem of metabolic dysregulation in AMD [116]. They showed that RPE cells isolated from deceased AMD donors showed an increased expression of *PARP2* (poly(ADP-ribose) polymerase 2), decreased NAD^+^, dysfunctional AMPK/SIRT1/PGC-1α pathway as compared with RPE cells isolated from non-AMD eyes. AMD eyes also showed overactive mTOR pathways. To directly investigate the metabolomics dysregulation, they analyzed untargeted metabolomics and lipidomic profiling of AMD RPE and observed altered levels of specific lipids and metabolites. Metabolomic and lipidomic changes were also observed in disorders similar to AMD, macular telangiectasia type 2, a bilateral eye disease with degeneration of glial cells and photoreceptors leading to central vision loss [117].

Golestaneh et al. showed earlier that the SIRT1/PGC-1α signaling is an important pathway in AMD pathogenesis as both proteins were repressed in this disease [118]. They used RPE cells that were obtained by the dedifferentiation of induced pluripotent stem cells (iPSCs) obtained by reprogramming of fibroblasts of AMD and non-AMD donors.

Insulin-like growth factor 1, a central protein in the nutrient signaling, may be implicated in AMD pathogenesis through its involvement in retinal degeneration and inflammation, playing an important role in AMD [119]. It was shown that insulin and IGFs were involved in the modulation of the equilibrium between growth and survival of retinal cells [120]. In the retina, IGF-1 is a potent proangiogenic factor and was detected in the neovascular membranes in wet AMD patients [121]. In addition, IGF-1 was reported to reduce apoptosis of photoreceptors in various retinitis pigmentosa models [122]. However, there are some controversies on the beneficial or detrimental effects of IGF-1 and its receptor on aging-related degenerative diseases (reviewed in [123]). A large case–control enrolling individuals from the Age-Related Eye Disease Study (AREDS) showed that the rs2872060 SNP in the gene encoding IGF-1 receptor was associated with the risk of both forms of advanced AMD: wet and dry [124]. In general, Arroba et al. using Igf1-deficient mice, showed that age-related malfunction of microglia induced a chronic low-grade inflammation [125]. Such a state favors retinal degeneration [119]. Castellino et al. showed that IGF-1 hematic levels were higher in wet AMD patients and patients with intermediate AMD as compared with controls without AMD, but no difference was observed between early AMD patients and controls [126]. Therefore, the role of IGF-1 in AMD pathogenesis may depend on the form of the disease (dry/wet) and its stage (early/intermediate/late). Some studies on AMD patients did not distinguish between those criteria, which may cause discrepancies between results obtained in different groups of patients.

### 3.2. Autophagy in AMD

The retina is the most metabolically active tissue in the human body, and so it produces relatively the highest level of ROS even in normal conditions [127]. This results in persistent oxidative stress in the retina. Photoreceptors, RPE cells and other structures of the retina can be affected by an elevated level of oxidative stress. Such stress results in an increased production of ROS and RNS that may damage biomolecules, including proteins. Damaged, oxidized proteins can be misfolded by conjugation with glutathione [128]. This induces UPR, in which damaged proteins are targeted by chaperones, but if this fails, soluble proteins are ubiquitinated and degraded in the proteasome, and molecular chaperones may assist this process [129]. Aging and neurodegeneration may result in dys- or non-functional proteasomal system, leading to accumulation of ubiquitinated proteins, which in turn are targeted by autophagy receptors, including p62/SQSTM1 (sequestosome 1) and LC3 [130].

We genetically modified mice through the induction of inactivating mutations in the nuclear factor, erythroid 2-related factor 2 (*NFE2L2*) and *PGC-1α* genes and these animals display features of dry AMD [131]. We observed an increased level of protein ubiquitination in both RPE and photoreceptors of these mice. PGC-1α is an important component of the aging stress response. In oxidative stress, NFE2L2 is translocated from the cytosol to the nucleus, where it stimulates the expression of several genes whose products are important for antioxidant defense [132]. Therefore, the model of dKO *PGC-1α*/*NFE2L2* mice is suitable for investigating the role of aging stress responses as it can be exploited at different animal ages.

Impaired autophagy has been suggested to play a role in AMD pathogenesis [133,134]. Accumulation of lipofuscin, resulted from disturbed autophagy, may induce activation of the NLRP3 (NLR family pyrin domain-containing 3) inflammasome, ultimately causing drusen formation and low-grade inflammation, typical for AMD [135]. However, the rationale for this conclusion has been partly questioned by Kosmidou et al. [136].

In stress conditions, there are many substrates for autophagy, but its high level may be associated with cellular death [137]. Therefore, autophagy may present two faces: pro-life and pro-death. The regulation of the interaction between these two pathways and their relationship to AMD pathogenesis is unknown. Mitter et al. suggested that oxidative stress may induce autophagy in RPE cells in two phases: an initial increase followed by a decrease to aggravate oxidative stress and prevent the accumulation of lipofuscin [138].

In searching for the mechanisms underlying autophagy impairment in AMD RPE cells, Jang et al. recently observed that activation of PARP1 by oxidative stress in ARPE-19 cells resulted in autophagy downregulation [139]. Moreover, these authors reported that an NAD^+^ precursor restored autophagy and protected mitochondria by the maintenance of SIRT1 activity in oxidative stress. Furthermore, a PARP1 inhibitor did not restore autophagy in SIRT1-depleted cells. These authors also used a mice model of AMD to confirm in vitro research. Altogether, their result showed that PARP1-dependent inhibition of SIRT1 activity affected autophagic survival of RPE cells, resulting in retinal degeneration, essential in AMD pathogenesis.

### 3.3. DDR in AMD

Persistent elevated oxidative stress in the AMD retina results in increased levels of ROS and RNS that damage DNA in retinal cells. Mitochondrial DNA can accommodate DNA damage to an increased extent than nDNA, as there are many copies of mtDNA in each mitochondrion and many mitochondria in each nucleated cell. High levels of damage in nDNA may induce cellular senescence, apoptosis or necrosis, while cells with a high level of mtDNA may survive as highly or persistently damaged mtDNA can be degraded [140,141]. However, the presence of multiple copies of damaged mtDNA may impact cellular functions and contribute to diseases [142]. As stated previously, ATM plays an important role in the aging stress response as it is a crucial DDR protein. Mauget-Faysse et al. enrolled patients with radiation-induced telangiectasia after radiotherapy for AMD and screened them for mutations in the *ATM* gene [143]. These authors observed that missense variants of *ATM* were associated with an AT-like phenotype and formation of retinal and choroidal vascular abnormalities.

We determined the extent of endogenous total and oxidative DNA damage, the susceptibility to exogenous genotoxic agents and the efficacy of DNA repair in lymphocytes of AMD patients and controls [144]. The cells from AMD patients displayed a greater extent of basal endogenous DNA damage. The extent of oxidative modification to DNA bases was greater in AMD patients than in the controls, as checked by DNA repair enzymes NTH1 (endonuclease III-like protein 1) and Fpg (DNA-formamidopyrimidine glycosylase). Lymphocytes from AMD patients showed a higher susceptibility to hydrogen peroxide and UV light and repaired DNA damage induced by these factors less effectively than the cells from the controls. We postulated that the impaired efficacy of DDR expressed by decreased DNA repair might link with enhanced sensitivity of RPE cells to blue light, contributing to the pathogenesis of AMD. These studies also confirmed that AMD might be considered a systemic disease.

The presence of elevated DNA damage in AMD has been evidenced in other works, and several mechanisms may underline this effect, including the decreased activity of antioxidant enzymes, mutations in DDR proteins and impairment in some DDR signaling pathways [12,14]. Although the nature of this effect is not fully known, oxidative stress and resulting enhanced levels of ROS/RNS may contribute to it, not only by induction of DNA damage but also by damage to DDR proteins. In addition, the structure of chromatin in the nucleus and the mitochondrial membrane, essential for DDR, can be directly affected by ROS/RNA. In mitochondria, the sites of ROS production and mtDNA location overlap, so the repair of mtDNA may be especially difficult [145]. At present, we can only state that disturbed DDR may be associated with AMD, but we cannot answer the question, whether it is a consequence of the disease or belongs to its reasons.

Accumulation of mtDNA damage is observed in normal and premature aging [146]. This accumulation may result from increased production of ROS by aging mitochondria; damaged mtDNA is a source of damaged components of mtETC leaking more ROS—the classical vicious cycle [147]. However, mtDDR, impaired with aging, may also contribute to such accumulation. Lin et al. provided some evidence on impairing mtDDR in aging and AMD [148]. These authors compared mtDNA damage and repair in cultured human RPE primary cells obtained from the macula and peripheral regions of the retina and showed a greater extent of mtDNA damage in the former and an increase of the damage with the age of donors. They also observed a higher prevalence of heteroplasmic mutations in AMD than non-AMD eyes and concluded that impaired mtDDR might play a role in AMD pathogenesis.

Ferrington et al. showed that AMD donors with the high-risk allele for CFH displayed more mtDNA damage than donors with no risk allele [149]. Moreover, these authors did not observe such a greater extent of mtDNA damage in age-matched control donors carrying the risk allele, suggesting that such increased extent of mtDNA damage is not a direct consequence of the CFH risk variant. The authors attributed the increase in mtDNA damage to retinal changes associated with AMD onset and associated cellular disturbances in the retina.

### 3.4. mtQC in AMD

Mitochondria and mtQC are suggested to play an important role in AMD pathophysiology (reviewed in [11,150,151,152,153]). The involvement of mitochondrial DDR in AMD pathogenesis has been signalized in the previous section. Ferrington et al. and Golestaneh et al. independently reported reduced mitochondrial functions and ATP production in RPE cells isolated from human AMD eyes [154,155]. These and other studies suggest that AMD may be associated with an energetic crisis in the RPE [94]. However, other consequences of impaired mtQC in AMD were reported, including disrupted intracellular calcium homeostasis and nuclear-mitochondrial signaling [94,156].

Golestaneh et al. observed disintegrated mitochondria in AMD-RPE-iPSC cells and repressed PGC-1α, a central regulator of mitochondrial biogenesis [157].

As stated above, damage to mtDNA may lead to a vicious cycle, which may be avoided by the degradation of extensively damaged mtDNA. The mechanism of this effect is not fully known and can involve several factors, including DNA polymerase γ, MGME1 (mitochondrial genome maintenance exonuclease 1) and TFAM (reviewed in [158]). However, mtDNA need not be extensively damaged to initiate the vicious cycle, and the cycle may also be started by damaged mitochondria with integral mtDNA. Such mitochondria should be removed by mitophagy.

Mitochondrial damage in the RPE is considered as a trigger to events leading to degradation RPE and photoreceptors, essential for AMD [11]. Brown et al. conditionally deleted the gene encoding the mitochondrial antioxidant enzyme, manganese superoxide dismutase, in the RPE of BALB/cJ mice that are established models of AMD [159]. They observed reduced RPE functions with age in mice with knockout, as well as changed the structure and functions of their RPE mitochondria. These changes included severe disruption of photoreceptor mitochondria, mitochondria fragmentation and alterations in the metabolism of RPE and photoreceptors, which may substantially contribute to the mechanisms of retinal degeneration in AMD.

Senescence and aging of mitochondria are interconnected, and mitochondrial dysfunctions may be a significant cause of senescence, and senescent cells contribute to senescence-associated mitochondrial dysfunction (SAMD) [151]. Therefore, SAMD may represent a mechanism of the involvement of impaired mtQC in AMD pathogenesis as senescence of RPE cells plays an important role in AMD pathogenesis (reviewed in [131]).

## 4. Conclusions and Perspectives

The insulin/IGF-1, mTOR, AMPK and sirtuin signaling pathways join cellular and organismal homeostasis through the regulation of nutrient sensing and stress response mechanisms, including proteostasis, mtQC and genome maintenance. The coordinated action of these systems is mainly responsible for fitness in the first stages of life, but their impairment may contribute to aging and age-related diseases. On the other hand, aging decreases the efficacy of many processes of the stress response, and it itself is a stressor. The aging stress response includes pathways that are modulated by the aging process. Therefore, the outcome of a balance between stress response and aging-related changes imposed on this response determines the fitness of an organism and its biological aging. It is also responsible for premature aging and age-related diseases. The aging stress response can be understood as adaptive homeostasis. Adaptive to aging.

Age-related macular degeneration is a disease with age as its main pathogenesis factor. Although different authors and authorities provide different age limits from which the persistence of this disease is common—55, 60 or 65 years, it is known that many people of an advanced age do not suffer from AMD and other age-related diseases, including cardiac diseases and cancer. Two factors seem to be primarily responsible for such a situation: external environment/lifestyle and genetic constitution understood in a broad sense, including not only gene variants of known AMD susceptibility, but also variants to face external and internal environment challenged by the aging process. The aging stress response is, therefore, almost by definition, an important factor of AMD pathogenesis. Disturbed autophagy, DDR and mtQC are reported in AMD patients and various cellular and animal models. In addition, the role of the insulin/IGF-1 pathway may be predominant in AMD pathogenesis related to the aging stress response. Moreover, SIRT1, mTOR, IGF-1, PGC-1α and other essential proteins of the aging stress response were shown to be disturbed in AMD patients and cellular/animal models of AMD.

It is worth noting that PGC-1α is involved in nearly all, if not all, pathways of the aging stress response (Figure 6). As we reasoned in our previous work, this protein may be crucial in AMD pathogenesis due to its protective role against senescence of RPE cells in oxidative stress [10]. Mice with the manipulated expression of the *PGC-1a* gene were shown to mimic human retinal diseases, including AMD [131,158,159,160,161].

In perspective, further studies are needed on the aging stress response to provide more details and build an integrated network of responses and maintenance pathways, which are to face external and internal challenges under the pressure of aging. In AMD patients, various aspects of aging stress response should be investigated with a special interest in PGC-1α, which plays a role in many pathways of the response. Induced pluripotent stem cells may be produced from AMD patients with defects in the aging stress response or with engineered such defects and then dedifferentiate into RPE cells.

## Figures and Tables

**Figure 1 ijms-21-08840-f001:**
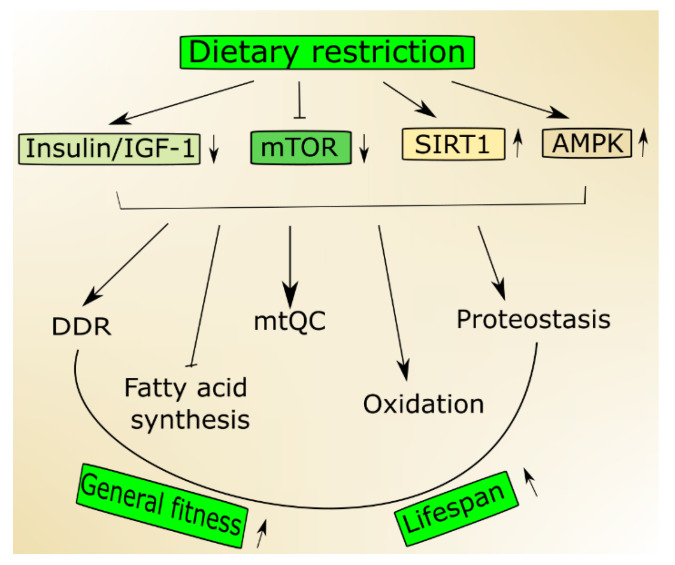
Nutrition signaling includes proteins and signaling pathways important in aging. Dietary restriction activates 5’AMP-activated protein kinase (AMPK) and sirtuin 1 (SIRT1) but inhibits the insulin growth factor 1 (insulin/IGF-1) and mechanistic target of rapamycin (mTOR) that stimulate genome maintenance by improving DNA damage response (DDR) and mitochondrial functions and eliminate damaged mitochondria through mitochondrial quality control (mtQC). Both mTOR decrease and SIRT1 increase may contribute to improved proteostasis, and AMPK supports the change from fatty acid synthesis to oxidation. Altogether these actions may result in an increase of biological fitness and extension of the lifespan of an organism.

**Figure 2 ijms-21-08840-f002:**
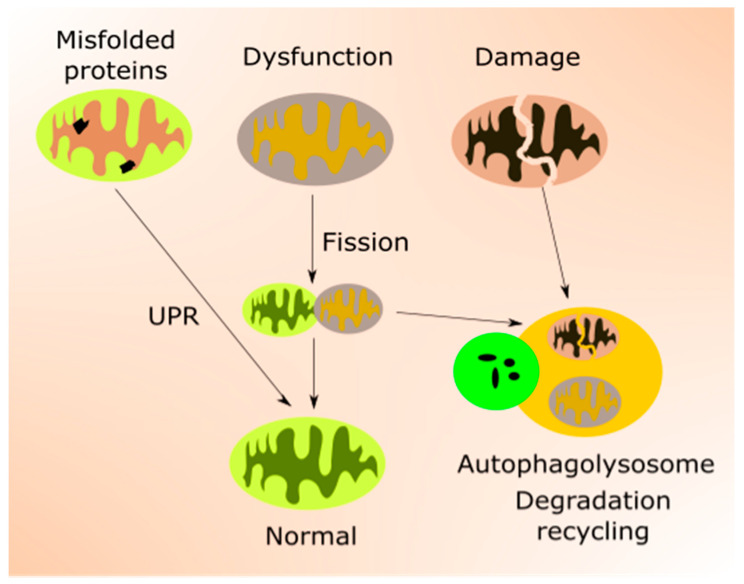
Mitochondrial quality control (mtQC) pathways repair and eliminate impaired mitochondria. Misfolded or unfolded proteins in mitochondria can be refolded by unfolded protein response (UPR). Dysfunctional mitochondria can be sequestered by fission and structurally damaged mitochondria eliminated by mitophagy, a specialized form of autophagy. Mitochondrial biogenesis, an important element of mtQC, is not presented in this figure.

**Figure 3 ijms-21-08840-f003:**
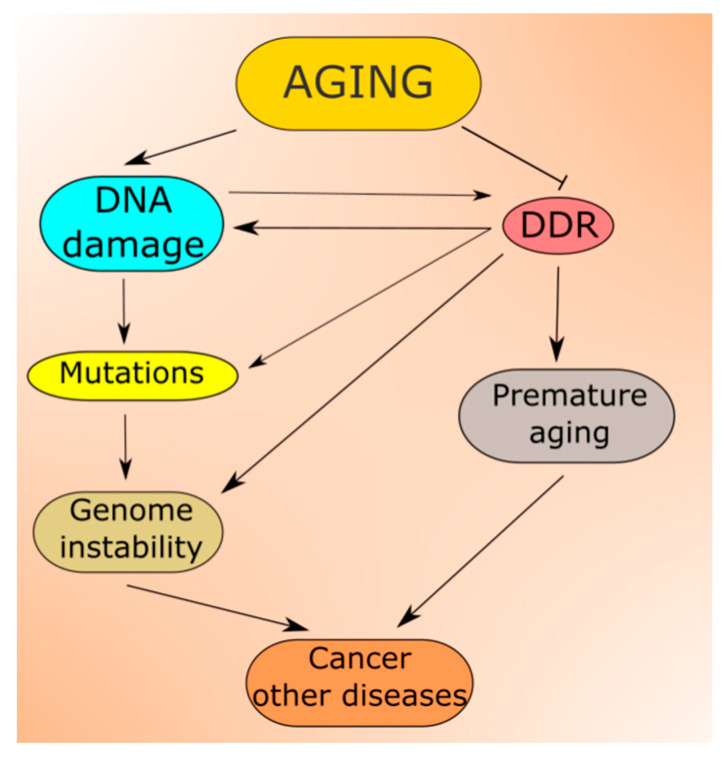
Aging is associated with increased levels of DNA damage, which can be changed into mutations contributing to increased genomic instability, a prerequisite in cancer and many other serious diseases. In normal conditions, DNA damage induces DNA damage response (DDR), which activates cellular pathways to repair the damage, tolerate it or induce programmed cell death. However, aging also affects these pathways lowering the efficacy and accuracy of DDR, increasing the extent of non-repaired and/or misrepaired DNA damage. In fact, faulty DDR may accelerate aging leading to premature aging and progeria syndromes.

**Figure 4 ijms-21-08840-f004:**
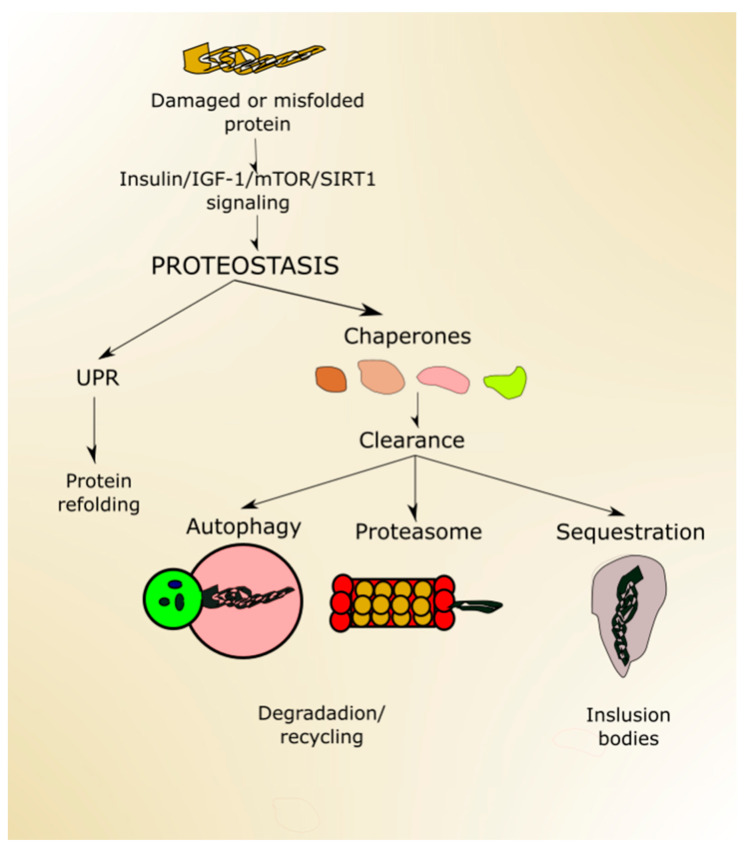
Proteostasis ensures protein homeostasis and is regulated by proteins and signaling pathways playing a role in aging, including (insulin-like growth factor 1 (insulin/IGF-1), mechanistic target of rapamycin (mTOR) and sirtuin 1 (SIRT1). Unfolded or misfolded proteins can be targeted by unfolded protein response (UPR), and damaged proteins can be designated to degradation in the proteasome or autophagy or sequestered in inclusion bodies. All these processes may be assisted by chaperones.

**Figure 5 ijms-21-08840-f005:**
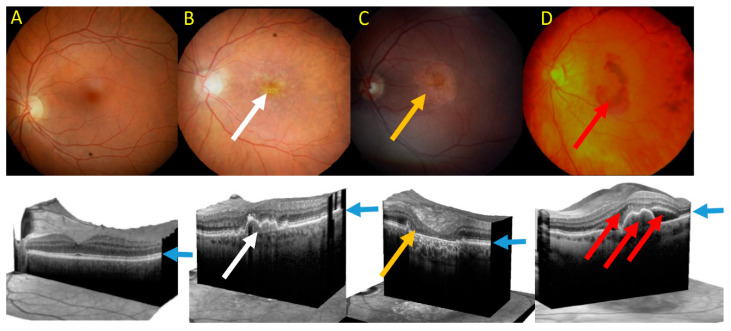
Fundus photograph of healthy macula (**A**) and macula of age-related macular degeneration (AMD) in its early form with several drusen deposits (white arrow) (**B**), geographic atrophy (GA) with typical GA lesions (yellow arrow) (**C**) and wet form with subretinal pigment epithelium (RPE) and subretinal and intraretinal fluids and hemorrhage (red arrow) (**D**). Lower panel—corresponding images of optical coherence tomography. Blue arrows point at the RPE layer.

**Figure 6 ijms-21-08840-f006:**
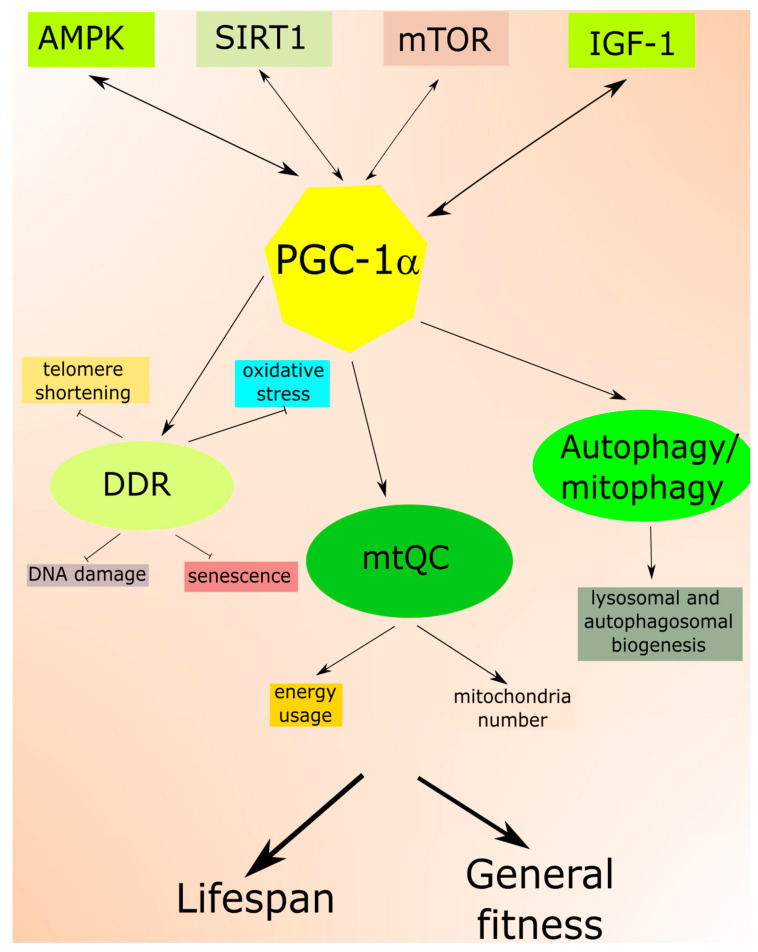
The central role peroxisome proliferation-activated receptor coactivator 1 alpha (PGC-1α) in aging stress response. PGC-1α interacts with proteins playing an important role in aging regulation: 5’AMP-activated protein kinase (AMPK), sirtuin 1 (SIRT1), mechanistic target of rapamycin (mTOR) and insulin-like growth factor 1 (IGF-1). These interactions result in the involvement of PGC-1α in the essential aging stress response pathways: DNA damage response (DDR), mitochondrial quality control (mtQC) and autophagy/mitophagy. PGC-1α is a primary regulator of antioxidant defense, and in this way, it can ameliorate DNA damage, telomere shortening and oxidative stress- and DNA damage-related senescence. PGC-1α controls mitochondrial response to stress associated with low energy allowing mitochondria to adjust their energy usage and number. PGC-1α may also contribute to the biogenesis of lysosomes and autophagosomes, essential for autophagy and mitophagy. These effects may extend lifespan and improve general fitness.

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
