# Peer review of "The Aging Stress Response and Its Implication for AMD Pathogenesis"

_ijms, 2020, doi:10.3390/ijms21228840_

Round 1
Reviewer 1 Report
It was a pleasure to work through the manuscript entitled “The Aging Stress Response and its Implication for AMD Pathogenesis” by J. Blasiak and colleagues. This narrative review brings together a large number of pieces of the puzzle in an attempt to provide a picture of the current knowledge about physiological and aging-associated stress response pathways and their contribution to pathogenesis and progression of AMD. The topic in this rapidly evolving field is well chosen, probably more for an academic than a broad clinical field of readers.
The authors have significantly contributed to this field. Their profound knowledge regarding the 4 major processes involved in the attempts to cope with the with increasing age decreasingly successful stressors is well perceived, whereas the manuscript in its current state serves many pieces of the puzzle, but does not sufficiently set them into the context to provide the composite picture that could result from bringing together all these pieces of knowledge. The linguistic performance clearly contributes to this and deserves improvement. Further critics in detail:
The abstract is – compared to the body of the manuscript – not sufficiently informative. A complete re-writing is suggested in order to guide the reader into the exciting field that they open.
A lack of references to several of the statements deserves attention though the long and qualified list of references indicates an exhaustive search of literature. Examples are lines 58, 60, 61, 69, 78, 135, 179, etc and namely the source references for all figures.
Figure 1: Insulin/IGF-1 decreases in age. The text (line 91) seems to contradict that blockage of insulin/IGF-1 goes along with general fitness and an improved life span.
It would be great to add a scheme presenting the interrelation between PGC-1a, stress, age and lifespan.
Line 156: define or elaborate “an appropriate stimulus”.
Line 294: change to “RPE cell atrophy”.
Line 294: what do the authors wish to say here with “the RPE is integral” ?
Lines 310-311: To my understanding, amyloid-beta has no link to aging stress response. It might be fruitful to spend few more lines to position amyloid-beta into this context.
Lines 320-1: There are clear concepts on how mediterranian diet could decrease early AMD prevalence. Please spend a few words on the presumed mechanism which fits well in the context.
Lines 322-4: The same accounts for fatty acids: Please provide the link to he aforementioned here, what is the pathophysiology behind?
Lines 362-5: Again, this interesting information is a little lost and should be linked to the aforementioned.
Lines 409-10: I absolutely agree that AMD has to be considered the local manifestation of a systemic…, probably not disease, but genetic disposition in the context of environmental influences. At the end, all is in our mighty genes. This would well provide the link between chapters 2 and 3 and could be positioned more prominently before chapter 2 or between the 2 chapters.
In the conclusion, I miss a sentence about the predominant role of Insulin/IGF-1.
In the perspective, I think that current knowledge has not progressed far enough to allow a link to possible new treatment options. More informative would be where to set the focus of research for the next decade.
I look forward to read the revision of this interesting manuscript.
Author Response
It was a pleasure to work through the manuscript entitled “The Aging Stress Response and its Implication for AMD Pathogenesis” by J. Blasiak and colleagues. This narrative review brings together a large number of pieces of the puzzle in an attempt to provide a picture of the current knowledge about physiological and aging-associated stress response pathways and their contribution to pathogenesis and progression of AMD. The topic in this rapidly evolving field is well chosen, probably more for an academic than a broad clinical field of readers.
Comment: The authors have significantly contributed to this field. Their profound knowledge regarding the 4 major processes involved in the attempts to cope with the with increasing age decreasingly successful stressors is well perceived, whereas the manuscript in its current state serves many pieces of the puzzle, but does not sufficiently set them into the context to provide the composite picture that could result from bringing together all these pieces of knowledge.
Answer: We tried to improve the internal logic and consistency of the manuscript.
Comment: The linguistic performance clearly contributes to this and deserves improvement.
Answer: We have tried to do our best to improve the style and correct all linguistic errors in the manuscript.
Further critics in detail:
Comment: The abstract is – compared to the body of the manuscript – not sufficiently informative. A complete re-writing is suggested in order to guide the reader into the exciting field that they open.
Answer: We have completely rewritten the abstract that now looks like:
Aging induces several stress response pathways to counterbalance detrimental changes associated with this process. These pathways include nutrient signaling, proteostasis, mitochondrial quality control and DNA damage response. At the cellular level, these pathways are controlled by evolutionary conserved signaling molecules, such as 5'AMP-activated protein kinase (AMPK), mechanistic target of rapamycin (mTOR), insulin/insulin like growth factor 1 (IGF-1) and sirtuins, including SIRT1. Peroxisome proliferation-activated receptor coactivator 1 alpha (PGC-1α), encoded by the PPARGC1A gene, playing an important role in antioxidant defense and mitochondrial biogenesis, may interact with these molecules influencing lifespan and general fitness. Perturbation in the aging stress response my lead to aging-related disorders, including age-related macular degeneration (AMD), the main reason of vision loss in the elderly. This is supported by studies showing an important role of disturbances in mitochondrial metabolism, DDR, and autophagy in AMD pathogenesis. Also, disturbed expression of PGC-1α was shown to associate with AMD. Therefore, the aging stress response may be critical for AMD pathogenesis and further studies are needed to precisely determine mechanisms underlying its role in AMD. These studies can include research on retinal cells produced from pluripotent stem cells obtained from AMD donors with the mutations, either native or engineered, in the critical genes for the aging stress response, including AMPK, IGF1, MTOR, SIRT1 and PPARGC1A.
Comment: A lack of references to several of the statements deserves attention though the long and qualified list of references indicates an exhaustive search of literature. Examples are lines 58, 60, 61, 69, 78, 135, 179, etc and namely the source references for all figures.
Answer: The original submission included 155 references and that is why we refrained from adding references to support mostly direct conclusions and sentences presenting rather general descriptions. However, to follow the suggestion, we have made the following changes:
Line 58 in the original submission: “These proteins should be removed and recycled by proteostasis, which dynamically regulates the proteome to keep it balanced and functional and is an essential pathway of the stress response [6].” – a reference has been added.
Line 60: “A damage to DNA induces DDR, another stress response component [2].” – a reference has been added
Line 61: “However, oxidative stress and increased ROS/RNS levels are associated with several other AMD risk factors, including improper diet, tobacco smoking, blue light exposure and others [3].”
Line 69: We have changed the sentence:
“There are substantial differences in everyday life between young and older adults, such as nutrient intake and physical activity that may influence AMD susceptibility, but the aging stress response seems to be a prime candidate to distinguish AMD susceptibility of younger and older adults.”
into the following fragment (a reasoning instead a reference in the second sentence):
“There are substantial differences in everyday life between young and older adults, such as nutrient intake and physical activity that may influence AMD susceptibility [9]. The aging stress response seems to be a prime candidate to distinguish AMD susceptibility of younger and older adults as it combats many AMD risk factors and is declined with aging.”
Line 78: “However, the causal relationship between the biological changes that occur with time (biological aging) and metrical (chronological) aging is far from understanding [10].” – a reference has been added
Line 135: “Moreover, the most pronounced antioxidant effect was observed for mtDNA, suggesting that aging might be linked with the production of ROS in mitochondria [37].” – a reference has been added
Line 178: “Moreover, not only DNA, but other cellular macromolecules undergo damage in aging [53].”
Figures: We were inspired by several sources of information in figures preparation. The most closely related references are mentioned in the text concerning a specific figure. e.g. when the Figure 1 is cited, the reference of 21 is also cited, Figure 2 – reference 44 and so on.
Additional references:
- Vilchez, D.; Saez, I.; Dillin, A. The role of protein clearance mechanisms in organismal ageing and age-related diseases. Nat Commun 2014, 5, 5659, doi:10.1038/ncomms6659.
- Gorgoulis, V.G.; Pefani, D.-E.; Pateras, I.S.; Trougakos, I.P. Integrating the DNA damage and protein stress responses during cancer development and treatment. The Journal of pathology 2018, 246, 12-40, doi:10.1002/path.5097.
- Jarrett, S.G.; Boulton, M.E. Consequences of oxidative stress in age-related macular degeneration. Molecular aspects of medicine 2012, 33, 399-417, doi:10.1016/j.mam.2012.03.009.
- Carneiro, Â.; Andrade, J.P. Nutritional and Lifestyle Interventions for Age-Related Macular Degeneration: A Review. Oxidative medicine and cellular longevity 2017, 2017, 6469138-6469138, doi:10.1155/2017/6469138.
- Ferrucci, L.; Gonzalez-Freire, M.; Fabbri, E.; Simonsick, E.; Tanaka, T.; Moore, Z.; Salimi, S.; Sierra, F.; de Cabo, R. Measuring biological aging in humans: A quest. Aging cell 2020, 19, e13080-e13080, doi:10.1111/acel.13080.
37 Giorgi, C.; Marchi, S.; Simoes, I.C.M.; Ren, Z.; Morciano, G.; Perrone, M.; Patalas-Krawczyk, P.; Borchard, S.; Jędrak, P.; Pierzynowska, K., et al. Mitochondria and Reactive Oxygen Species in Aging and Age-Related Diseases. International review of cell and molecular biology 2018, 340, 209-344, doi:10.1016/bs.ircmb.2018.05.006.
- Campisi, J.; Vijg, J. Does damage to DNA and other macromolecules play a role in aging? If so, how? J Gerontol A Biol Sci Med Sci 2009, 64, 175-178, doi:10.1093/gerona/gln065.
Comment: Figure 1: Insulin/IGF-1 decreases in age. The text (line 91) seems to contradict that blockage of insulin/IGF-1 goes along with general fitness and an improved life span.
Answer: We have changed Figure 1 accordingly.
Comment: It would be great to add a scheme presenting the interrelation between PGC-1a, stress, age and lifespan.
Answer: We have added Figure 6 .
Comment: Line 156: define or elaborate “an appropriate stimulus”.
Answer: We have changed the sentences:
“An appropriate stimulus upregulates transcription factors activating nuclear genes that code for mitochondrial proteins, including transcription factors. These factors, with mitochondrial transcription factor A (TFAM), are transported to mitochondria and target mtDNA.”
into:
“Mitochondrial transcription factors, including mitochondrial transcription factor A (TFAM), are encoded by nuclear genes and transported to mitochondria to target mtDNA.”
Comment: Line 294: change to “RPE cell atrophy”.
Answer: We have done so.
Comment: Line 294: what do the authors wish to say here with “the RPE is integral”?
Answer: We have removed “is integral”.
Comment: Lines 310-311: To my understanding, amyloid-beta has no link to aging stress response. It might be fruitful to spend few more lines to position amyloid-beta into this context.
Answer: We are not sure whether we understand this comment. We have added the following fragment:
“Many studies report an enhanced release of the brain beta amyloid 1-42 (Aβ) in stress conditions [104]. Moreover, the production and clearance of Aβ is reported to depend on stress [105]. Therefore, the deposition of Aβ might result from a decline of the stress response with aging. In summary, amyloid beta deposits observed in AMD retinas may be associated with declined aging stress response.”
with the following new references:
- Morgese, M.G.; Schiavone, S.; Trabace, L. Emerging role of amyloid beta in stress response: Implication for depression and diabetes. European journal of pharmacology 2017, 817, 22-29, doi:10.1016/j.ejphar.2017.08.031.
- Butterfield, D.A.; Boyd-Kimball, D. Oxidative Stress, Amyloid-β Peptide, and Altered Key Molecular Pathways in the Pathogenesis and Progression of Alzheimer’s Disease. Journal of Alzheimer's Disease 2018, 62, 1345-1367, doi:10.3233/JAD-170543.
Comment: Lines 320-1: There are clear concepts on how mediterranian diet could decrease early AMD prevalence. Please spend a few words on the presumed mechanism which fits well in the context.
Answer: To address this point we have added some information on AMD genetic risk factors in 3. Age-related macular degeneration – a disease of aging stress response section:
“To date, more than half of the AMD genomic heritability can be explained by both common and rare genetic variants in 34 genetic loci, which are mostly involved in the complement system, extracellular matrix remodeling, and lipid metabolism [106]. A common variant (rs1061170) in the complement factor H (CFH) has been identified by several research group to strongly influence AMD risk [107]. The age-related maculopathy susceptibility 2 and high-temperature requirement A serine peptidase 1 (ARMS2/HTRA1) is another locus reported to be strongly involved in AMD pathogenesis [108].”
with the following references:
- Fritsche, L.G.; Igl, W.; Bailey, J.N.; Grassmann, F.; Sengupta, S.; Bragg-Gresham, J.L.; Burdon, K.P.; Hebbring, S.J.; Wen, C.; Gorski, M., et al. A large genome-wide association study of age-related macular degeneration highlights contributions of rare and common variants. Nature genetics 2016, 48, 134-143, doi:10.1038/ng.3448.
- Park, D.H.; Connor, K.M.; Lambris, J.D. The Challenges and Promise of Complement Therapeutics for Ocular Diseases. Front Immunol 2019, 10, doi:10.3389/fimmu.2019.01007.
- Grassmann, F.; Heid, I.M.; Weber, B.H.F.; International, A.M.D.G.C. Recombinant Haplotypes Narrow the ARMS2/HTRA1 Association Signal for Age-Related Macular Degeneration. Genetics 2017, 205, 919-924, doi:10.1534/genetics.116.195966.
We have added the following fragment to the first paragraph in the 3.1 Nutrient signaling in AMD section:
“Adherence to such a diet was positively correlated with a lower incidence of AMD in several other studies [9,112-114]. A Mediterranean-type diet is rich in plant foods (fruits, nuts, legumes and cereals) and fish, with olive oil as the dominating fat and moderate amount of wine and low amount of red meat and poultry. It was generally observed that the closer adherence to this kind of diet slower the progression of early AMD to its advanced form and to larger drusen and this relationship was modulated by the CFH genotype suggesting that the biological mechanism behind the protective action of Mediterranean-type diet is involved with the complement system.”
with the following references:
- Keenan, T.D.; Agrón, E.; Mares, J.; Clemons, T.E.; van Asten, F.; Swaroop, A.; Chew, E.Y. Adherence to the Mediterranean Diet and Progression to Late Age-Related Macular Degeneration in the Age-Related Eye Disease Studies 1 and 2. Ophthalmology 2020, 127, 1515-1528, doi:10.1016/j.ophtha.2020.04.030.
- Merle, B.M.J.; Silver, R.E.; Rosner, B.; Seddon, J.M. Adherence to a Mediterranean diet, genetic susceptibility, and progression to advanced macular degeneration: a prospective cohort study. The American journal of clinical nutrition 2015, 102, 1196-1206, doi:10.3945/ajcn.115.111047.
- Raimundo, M.; Mira, F.; Cachulo, M.D.L.; Barreto, P.; Ribeiro, L.; Farinha, C.; Laíns, I.; Nunes, S.; Alves, D.; Figueira, J., et al. Adherence to a Mediterranean diet, lifestyle and age-related macular degeneration: the Coimbra Eye Study - report 3. Acta Ophthalmol 2018, 96, e926-e932, doi:10.1111/aos.13775.
Comment: Lines 322-4: The same accounts for fatty acids: Please provide the link to he aforementioned here, what is the pathophysiology behind?
Answer: We have added the following fragment to continue the paragraph on fatty acids in AMD:
“Although biochemical and epidemiological studies suggest a high potential of fatty acids in AMD, the molecular mechanism of their involvement in AMD pathogenesis and specifically drusen formation ids poorly known [116]. Despite the promising results from dietary interventions with fatty acids, it is still not clear whether the fatty acids that enter the body through the diet or their counterpart produced locally by specific cells are to be the target for dietary intervention. Fatty acids are consumed along with other nutrients that could affect lipid metabolism, which likely results that the Mediterranean-type diet was reported to be more effective than a diet supplemented with just lipids [117].”
with the references:
- Skowronska-Krawczyk, D.; Chao, D.L. Long-Chain Polyunsaturated Fatty Acids and Age-Related Macular Degeneration. Advances in experimental medicine and biology 2019, 1185, 39-43, doi:10.1007/978-3-030-27378-1_7.
- Hogg, R.E.; Woodside, J.V.; McGrath, A.; Young, I.S.; Vioque, J.L.; Chakravarthy, U.; de Jong, P.T.; Rahu, M.; Seland, J.; Soubrane, G., et al. Mediterranean Diet Score and Its Association with Age-Related Macular Degeneration: The European Eye Study. Ophthalmology 2017, 124, 82-89, doi:10.1016/j.ophtha.2016.09.019.
Comment: Lines 362-5: Again, this interesting information is a little lost and should be linked to the aforementioned.
Answer: We have added the following fragment to continue that paragraph”
“PGC-1α is an important component of the aging stress response. In oxidative stress, NFE2L2 is translocated from the cytosol to the nucleus, where it stimulates the expression of several genes whose products are important for antioxidant defense [132]. Therefore, the model of dKO PGC-1a/NFE2L2 mice is suitable for the investigation the role of aging stress response as it can be exploited at different animal ages.”
with the reference:
- Hyttinen, J.M.T.; Kannan, R.; Felszeghy, S.; Niittykoski, M.; Salminen, A.; Kaarniranta, K. The Regulation of NFE2L2 (NRF2) Signalling and Epithelial-to-Mesenchymal Transition in Age-Related Macular Degeneration Pathology. International journal of molecular sciences 2019, 20, doi:10.3390/ijms20225800.
Comment: Lines 409-10: I absolutely agree that AMD has to be considered the local manifestation of a systemic…, probably not disease, but genetic disposition in the context of environmental influences. At the end, all is in our mighty genes. This would well provide the link between chapters 2 and 3 and could be positioned more prominently before chapter 2 or between the 2 chapters.
Answer: We are sorry, but we are afraid that we do not entirely follow the Reviewer’s reasoning. Several works, including our recent one (PMID: 31633290), support the suggestion that AMD, at least in its wet form, can be considered as a systemic disease in which not only vision loss, but general mortality might be considered as its direct consequences. We think that elaborating this subject in this manuscript could disturb its logical chain we tried to build and lead to its overload with data and references loosely associated with is main subject. Furthermore, AMD in this manuscript is mentioned in Abstract and Introduction, but in fact it is introduced in 3 section. Presenting some additional information at the end of the Introduction section would suggest that this information would be developed in the subsequent sections. Likewise, developing a subsection on systemic character of AMD between sections 2 and 3 would suggest that this subject would be developed. As we said, this would certainly lead to overloading of the manuscript with data and references not necessarily directly linked to its main subject.
Comment: In the conclusion, I miss a sentence about the predominant role of Insulin/IGF-1.
Answer: We have added the following sentence to the last section:
“Also, the role of the insulin/IGF-1 pathway may be predominant in AMD pathogenesis related to aging stress response.”
Comment: In the perspective, I think that current knowledge has not progressed far enough to allow a link to possible new treatment options. More informative would be where to set the focus of research for the next decade.
Answer: We have removed the last sentence in the manuscript, although it was a conditional clause, assuming that the progress in studies on the aging stress response in AMD with pluripotent stem cells that can be seen as a perspective to the next decade, would be sufficient to think about new therapeutic options related to these studies.
Comment: I look forward to read the revision of this interesting manuscript.
Answer: Thank you very much for essential, substantial, and constructive comments.
Reviewer 2 Report
Thank you for the opportunity to review this manuscript entitled ‘The Aging Stress Response and its Implication for AMD Pathogenesis’ by J. Blasiak and co-authors. In the present study, authors summarized current data on the contribution of aging to pathogenesis AMD. This problem is very relevant due to the pace of population aging in the world. The manuscript is well written and logical and presents review of current scientific evidence appropriate for publication in Antioxidants journal without changes.Author Response
Thank you for the opportunity to review this manuscript entitled ‘The Aging Stress Response and its Implication for AMD Pathogenesis’ by J. Blasiak and co-authors. In the present study, authors summarized current data on the contribution of aging to pathogenesis AMD. This problem is very relevant due to the pace of population aging in the world. The manuscript is well written and logical and presents review of current scientific evidence appropriate for publication in Antioxidants journal without changes.
Thank you.
Reviewer 3 Report
In the first half of the manuscript, the authors reviewed the involvement of several cellular stress responses, including nutrient sensing, mitochondria, DNA damage response and proteostasis, in general aging. Then in the second half of the paper the authors further explored the roles of these processes in age-related macular degeneration (AMD). The review has covered a broad range of topics and has many updated references, some of which are based on the authors’ own work. There are important statements, summaries, and conclusions that can be of interest to the readers. Because genetic factors are heavily involved in the etiology of AMD, the authors are recommended to add additional discussions on how the aging stress response can interact with the genetic predispositions of AMD. Other minor comments include:
Page 2, line 48-51. Statements on the number of US patients with AMD are wrong. The “196 million” and “288 million” were copied from reference 4, and those are estimates for the entire world population, not only US.
Page 2, line 52. The “lack of effective therapy” is not applicable to wet AMD
Page 2, line 62, UV light does not effectively penetrate beyond the lens and is not known as a key environment factor of AMD
Page 8 to page 9, introduction of AMD, and also Fig. 5, there is no discussion on geographic atrophy (GA). The presence of small, hard drusen is often seen in early AMD and GA is a distinct stage. For OCT image of panel C, the black arrows interfere with the view.
Page 9, line 294, “in wet AMD the RPE is integral”, the statement is either inaccurate or needs supporting reference.
Page 9, line 313, may consider using “micronutrients” instead of “nutritional interventions”
Page 10, 2nd paragraph, the discussions on IGF-1 did not adequately address the role of IGF-1 pathway in dry AMD, a condition that is more relevant to aging
Page 11, 1st paragraph. The discussions on mtDNA damage vs nDNA damage are likely to be inaccurate. First, findings from Ref 123 -127 mainly showed age-dependent increase in mtDNA damage, not AMD-associated increase. Second, especially for the non-dividing RPE cells, the presence of multiple copies of damaged mtDNA will likely impact cellular function, even though the affected cell(s) may not be in the stage of senescence, apoptosis or necrosis. The authors’ discussions likely undermine the importance of accumulated mtDNA damage
Page 11, lines 411-420, the cited references did not adequately support the role of disturbed DDR in AMD
Author Response
In the first half of the manuscript, the authors reviewed the involvement of several cellular stress responses, including nutrient sensing, mitochondria, DNA damage response and proteostasis, in general aging. Then in the second half of the paper the authors further explored the roles of these processes in age-related macular degeneration (AMD). The review has covered a broad range of topics and has many updated references, some of which are based on the authors’ own work. There are important statements, summaries, and conclusions that can be of interest to the readers.
Comment: Because genetic factors are heavily involved in the etiology of AMD, the authors are recommended to add additional discussions on how the aging stress response can interact with the genetic predispositions of AMD.
Answer: In some, if not many, cases, the reason(s) of association between a specific genotype and AMD occurrence is not known. Exploring the cases with known pathogenic mechanisms of specific gene variants in the context of aging stress response would cause overloading this manuscript with text, schemes, and references. This is not the only problem as the main difficulty follows from the fact that in a great majority of cases the genetic preference has not been determined. Instead we added some new information on the cases with the involvement of AMD-linked genetic predisposition. These include:
To address this point we have added some information on AMD genetic risk factors in 3. Age-related macular degeneration – a disease of aging stress response section:
“To date, more than half of the AMD genomic heritability can be explained by both common and rare genetic variants in 34 genetic loci, which are mostly involved in the complement system, extracellular matrix remodeling, and lipid metabolism [106]. A common variant (rs1061170) in the complement factor H (CFH) has been identified by several research group to strongly influence AMD risk [107]. The age-related maculopathy susceptibility 2 and high-temperature requirement A serine peptidase 1 (ARMS2/HTRA1) is another locus reported to be strongly involved in AMD pathogenesis [108].”
with the following references:
- Fritsche, L.G.; Igl, W.; Bailey, J.N.; Grassmann, F.; Sengupta, S.; Bragg-Gresham, J.L.; Burdon, K.P.; Hebbring, S.J.; Wen, C.; Gorski, M., et al. A large genome-wide association study of age-related macular degeneration highlights contributions of rare and common variants. Nature genetics 2016, 48, 134-143, doi:10.1038/ng.3448.
- Park, D.H.; Connor, K.M.; Lambris, J.D. The Challenges and Promise of Complement Therapeutics for Ocular Diseases. Front Immunol 2019, 10, doi:10.3389/fimmu.2019.01007.
- Grassmann, F.; Heid, I.M.; Weber, B.H.F.; International, A.M.D.G.C. Recombinant Haplotypes Narrow the ARMS2/HTRA1 Association Signal for Age-Related Macular Degeneration. Genetics 2017, 205, 919-924, doi:10.1534/genetics.116.195966.
We have added the following fragment to the first paragraph in the 3.1 Nutrient signaling in AMD section:
“Adherence to such a diet was positively correlated with a lower incidence of AMD in several other studies [9,112-114]. A Mediterranean-type diet is rich in plant foods (fruits, nuts, legumes and cereals) and fish, with olive oil as the dominating fat and moderate amount of wine and low amount of red meat and poultry. It was generally observed that the closer adherence to this kind of diet slower the progression of early AMD to its advanced form and to larger drusen and this relationship was modulated by the CFH genotype suggesting that the biological mechanism behind the protective action of Mediterranean-type diet is involved with the complement system.”
with the following references:
- Keenan, T.D.; Agrón, E.; Mares, J.; Clemons, T.E.; van Asten, F.; Swaroop, A.; Chew, E.Y. Adherence to the Mediterranean Diet and Progression to Late Age-Related Macular Degeneration in the Age-Related Eye Disease Studies 1 and 2. Ophthalmology 2020, 127, 1515-1528, doi:10.1016/j.ophtha.2020.04.030.
- Merle, B.M.J.; Silver, R.E.; Rosner, B.; Seddon, J.M. Adherence to a Mediterranean diet, genetic susceptibility, and progression to advanced macular degeneration: a prospective cohort study. The American journal of clinical nutrition 2015, 102, 1196-1206, doi:10.3945/ajcn.115.111047.
- Raimundo, M.; Mira, F.; Cachulo, M.D.L.; Barreto, P.; Ribeiro, L.; Farinha, C.; Laíns, I.; Nunes, S.; Alves, D.; Figueira, J., et al. Adherence to a Mediterranean diet, lifestyle and age-related macular degeneration: the Coimbra Eye Study - report 3. Acta Ophthalmol 2018, 96, e926-e932, doi:10.1111/aos.13775.
We have also added the following fragment to the 3.3. DDR in AMD section”
“Ferrington et al. showed that AMD donors with the high-risk allele for CFH displayed more mtDNA damage than donors with no risk allele [149]. Moreover, these authors did not observe such higher extent of mtDNA damage in age-matched control donors carrying the risk allele, suggesting that such increased extent of mtDNA damage is not a consequence suggests mt injury is not a direct consequence of the CFH risk variant. The authors attributed the increase in mtDNA damage to retinal changes associated with AMD onset and associated cellular disturbances in the retina.”
with the reference:
- Ferrington, D.A.; Kapphahn, R.J.; Leary, M.M.; Atilano, S.R.; Terluk, M.R.; Karunadharma, P.; Chen, G.K.; Ratnapriya, R.; Swaroop, A.; Montezuma, S.R., et al. Increased retinal mtDNA damage in the CFH variant associated with age-related macular degeneration. Experimental eye research 2016, 145, 269-277, doi:10.1016/j.exer.2016.01.018.
Other minor comments include:
Comment: Page 2, line 48-51. Statements on the number of US patients with AMD are wrong. The “196 million” and “288 million” were copied from reference 4, and those are estimates for the entire world population, not only US.
Answer: Certainly! We have corrected that (please see below).
Comment: Page 2, line 52. The “lack of effective therapy” is not applicable to wet AMD
Answer: We have changed the fragment:
“Age-related macular degeneration (AMD) is the primary cause of legal blindness in the elderly in developed countries and its estimated prevalence in the US is 196 million in 2020 and is projected to increase to 288 million by 2040, but despite these numbers, no effective treatment is available, except some specific cases of this disease (reviewed in [4]). This lack of effective therapy is likely due to highly incomplete knowledge of mechanism(s) involved in AMD pathogenesis.”
into:
“Age-related macular degeneration (AMD) is the primary cause of legal blindness in the elderly in developed countries and its estimated prevalence in the world is 196 million in 2020 and is projected to increase to 288 million by 2040, but despite these numbers, no effective treatment is available, except some specific cases of the neovascular (wet) form of AMD (reviewed in [4]). However, the treatment in those cases is mainly aimed at the vision loss prevention. This limitation in effective therapeutic options is likely due to highly incomplete knowledge of mechanism(s) involved in AMD pathogenesis.”
Comment: Page 2, line 62, UV light does not effectively penetrate beyond the lens and is not known as a key environment factor of AMD
Answer: We have changed the expression:
“UV exposure”
into:
“blue light exposure”
Comment: Page 8 to page 9, introduction of AMD, and also Fig. 5, there is no discussion on geographic atrophy (GA). The presence of small, hard drusen is often seen in early AMD and GA is a distinct stage. For OCT image of panel C, the black arrows interfere with the view.
Answer: We have stressed that vision loss in dry AMD is caused by the progression of this form of the disease to GA in the 3. Age-related macular degeneration – a disease of aging stress response section.
Furthermore, we have changed Figure 5 (please see attached) and its legend, introducing GA.
Comment: Page 9, line 294, “in wet AMD the RPE is integral”, the statement is either inaccurate or needs supporting reference.
Answer: We have deleted “is integral”.
Comment: Page 9, line 313, may consider using “micronutrients” instead of “nutritional interventions”
Answer: We have changed as suggested.
Comment: Page 10, 2nd paragraph, the discussions on IGF-1 did not adequately address the role of IGF-1 pathway in dry AMD, a condition that is more relevant to aging
Answer: We are sorry, but we are unable to find information on the dry AMD-specific involvement of IGF-1 in AMD pathogenesis. We have added the following fragment to that paragraph:
“A large case-control enrolling individuals from the Age-Related Eye Disease Study (AREDS) showed that the rs2872060 SNP in the gene encoding IGF-1 receptor was associated with the risk of both forms of advanced AMD: wet and dry [126]. In general, Arroba et al. using Igf1-deficient mice showed that age-related malfunction of microglia induced a chronic low-grade inflammation [127]. Such state favors retinal degeneration [121]”
with the following references:
- Chiu, C.J.; Conley, Y.P.; Gorin, M.B.; Gensler, G.; Lai, C.Q.; Shang, F.; Taylor, A. Associations between genetic polymorphisms of insulin-like growth factor axis genes and risk for age-related macular degeneration. Investigative ophthalmology & visual science 2011, 52, 9099-9107, doi:10.1167/iovs.11-7782.
- Arroba, A.I.; Rodríguez-de la Rosa, L.; Murillo-Cuesta, S.; Vaquero-Villanueva, L.; Hurlé, J.M.; Varela-Nieto, I.; Valverde Á, M. Autophagy resolves early retinal inflammation in Igf1-deficient mice. Disease models & mechanisms 2016, 9, 965-974, doi:10.1242/dmm.026344.
- Kauppinen, A.; Paterno, J.J.; Blasiak, J.; Salminen, A.; Kaarniranta, K. Inflammation and its role in age-related macular degeneration. Cellular and molecular life sciences : CMLS 2016, 73, 1765-1786, doi:10.1007/s00018-016-2147-8.
Comment: Page 11, 1st paragraph. The discussions on mtDNA damage vs nDNA damage are likely to be inaccurate. First, findings from Ref 123 -127 mainly showed age-dependent increase in mtDNA damage, not AMD-associated increase.
Answer: Reference 123: Ferrington et al. Increased retinal mtDNA damage in the CFH variant associated with age-related macular degeneration. Exp Eye Res 2016 – the authors conclude that “Our results show that donors harboring the high risk allele for CFH had significantly more mtDNA damage […]. The absence of higher mtDNA damage in age-matched control donors carrying the risk allele suggests mt injury is not a direct consequence of the CFH risk variant. Rather, retinal changes associated with the onset of disease coupled with the presence of the risk allele create cellular conditions conducive for accelerated mtDNA damage to occur.” We think that this justifies considering this work as supporting the thesis on increased mtDNA damage in AMD. However, that work does not directly suggest that mtDNA in AMD is more susceptible than its nuclear counterpart (nDNA).
Reference 124: Godley et al. Blue light induces mitochondrial DNA damage and free radical production in epithelial cells. J Biol Chem 2005 – we understand that work as showing that blue light induce mtDNA damage contributing to retinal pathologies, including AMD, but also aging per se. Moreover, the authors conclude on an increased vulnerability of mtDNA as compared to nDNA.
Reference 125: Jarrett et al. Mitochondrial DNA damage and its potential role in retinal degeneration. PRER 2008 - the authors conclude that “Mitochondrial dysfunction and DNA damage is associated with AMD”. Moreover, authors’ consideration between differences in vulnerability of mtDNA and nDNA to damage may suggest that mtDNA is more damaged that nDNA in AMD.
Reference 126: Karunadharma et al. Mitochondrial DNA damage as a potential mechanism for age-related macular degeneration. Invest Ophthalmol Vis Sci 2010 – the authors conclude “mtDNA is preferentially damaged with AMD progression” and age-specific damage to mtDNA has a different character. Moreover, the authors conclude that “mtDNA accumulated more lesions than did two nuclear genes, with total damage of the mt genome estimated to be eight times higher” suggesting that mtDNA is more prone to DNA damage than nDNA.
Reference 127: Liang and Godley. Oxidative stress-induced mitochondrial DNA damage in human retinal pigment epithelial cells: a possible mechanism for RPE aging and age-related macular degeneration. Exp Eye Res 2003 – the authors showed “that human RPE cells treated with H2O2 or rod outer segments resulted in preferential damage to mtDNA, but not nDNA” and that “susceptibility of mtDNA to oxidative damage in human RPE cells, together with the age-related decrease of cellular anti-oxidant system, provides the rationale for a mitochondria-based model of AMD”.
Therefore, we still consider that these references suggest that mtDNA damage may be associated with AMD. The question is: in what extend is this association age- and the disease-specific cannot be answered based on effects described in these works. Most of these references suggest a higher susceptibility of mtDNA to damage than nDNA in AMD.
Comment: Second, especially for the non-dividing RPE cells, the presence of multiple copies of damaged mtDNA will likely impact cellular function, even though the affected cell(s) may not be in the stage of senescence, apoptosis or necrosis.
Answer: It is obvious, and we do not question that – we have introduced a sentence on that subject (please see below).
Comment: The authors’ discussions likely undermine the importance of accumulated mtDNA damage.
Answer: It was not our intention to undermine the role of mtDNA damage in AMD pathogenesis. However, several reports seem to suggest that “in AMD mtDNA is more susceptible to damage than its nuclear counterpart” (please see above). We think that such opinion is not sufficiently evidenced, and we wanted to point out some methodological complications in comparison of the susceptibility of the nuclear and mitochondrial genomes to DNA-damaging factors. We keep on this opinion.
In summary – we have replaced the paragraph:
“Persistent elevated oxidative stress in AMD retina results in increased levels of ROS and RNS that damage DNA in retinal cells. Several studies suggest that mtDNA in AMD might be more damaged than its nuclear counterpart (nDNA) [123-127]. However, such thesis is not fully justified as we pointed out in our previous work [128]. In principle, mtDNA can accommodate DNA damage in an increased extent than nDNA as there are many copies of mtDNA in each mitochondrion and many mitochondria in each nucleated cell. High levels of damage in nDNA may induce cellular senescence, apoptosis or necrosis, while cells with a high level of mtDNA may survive as highly or persistently damaged mtDNA can be degraded [129,130]. Furthermore, DDR in mitochondria is significantly different from its nuclear counterpart [7].”
with:
“Persistent elevated oxidative stress in AMD retina results in increased levels of ROS and RNS that damage DNA in retinal cells. Mitochondrial DNA can accommodate DNA damage in an increased extent than nDNA as there are many copies of mtDNA in each mitochondrion and many mitochondria in each nucleated cell. High levels of damage in nDNA may induce cellular senescence, apoptosis or necrosis, while cells with a high level of mtDNA may survive as highly or persistently damaged mtDNA can be degraded [129,130]. However, the presence of multiple copies of damaged mtDNA may impact cellular functions and contribute to diseases [141].”
with the reference:
- Zhao, L.; Sumberaz, P. Mitochondrial DNA Damage: Prevalence, Biological Consequence, and Emerging Pathways. Chemical research in toxicology 2020, 33, 2491-2502, doi:10.1021/acs.chemrestox.0c00083.
Comment: Page 11, lines 411-420, the cited references did not adequately support the role of disturbed DDR in AMD.
Answer: We replaced references 133-138 in original version with two our reviews [12,14]:
- Blasiak, J.; Glowacki, S.; Kauppinen, A.; Kaarniranta, K. Mitochondrial and nuclear DNA damage and repair in age-related macular degeneration. International journal of molecular sciences 2013, 14, 2996-3010, doi:10.3390/ijms14022996.
- Hyttinen, J.M.T.; Błasiak, J.; Niittykoski, M.; Kinnunen, K.; Kauppinen, A.; Salminen, A.; Kaarniranta, K. DNA damage response and autophagy in the degeneration of retinal pigment epithelial cells-Implications for age-related macular degeneration (AMD). Ageing research reviews 2017, 36, 64-77, doi:10.1016/j.arr.2017.03.006.
Thank you very much for essential, substantial, and constructive comments.